# Experiences with implementation of continuous positive airway pressure for neonates and infants in low-resource settings: A scoping review

**Sara Dada**[1]*, **Henry Ashworth**[1,2], **Alina Sobitschka**[1,3], **Vanitha Raguveer**[1,4], **Rupam Sharma**[1,5], **Rebecca L. Hamilton**[6,7], **Thomas Burke**[1,2,8,9]

**1** Vayu Global Health Foundation Boston, Boston, Massachusetts, United States of America, **2** Harvard Medical School, Boston, Massachusetts, United States of America, **3** University of Göttingen, Göttingen, Germany, **4** University of Illinois College of Medicine, Chicago, Illinois, United States of America, **5** University of California Los Angeles Kern Medical Center, Bakersfield, California, United States of America, **6** Massachusetts General Hospital, Department of Anesthesiology, Boston, Massachusetts, United States of America, **7** Karolinska Institute, Department of Cell and Molecular Biology, Solna, Sweden, **8** Massachusetts General Hospital, Global Health Innovation Lab, Department of Emergency Medicine, Boston, Massachusetts, United States of America, **9** Harvard T. H. Chan School of Public Health, Boston, Massachusetts, United States of America

* sara.dada@ucdconnect.ie

## Abstract

### Background

Continuous positive airway pressure (CPAP) is the gold standard of care in providing non-invasive positive pressure support to neonates in respiratory distress in high-resource settings. While safety has been demonstrated in low-resource settings, there is a lack of knowledge on the barriers and facilitators to proper implementation.

### Objective

To identify and describe the barriers, facilitators, and priorities for future implementation of CPAP for neonates and infants in low-resource settings.

### Methods

A systematic search (database inception to March 6, 2020) was performed on MEDLINE, Embase, Web of Science, CINAHL, Global Health, and the WHO Global Index Medicus using PRISMA-ScR guidelines. Original research articles pertaining to implementation of CPAP devices in low-resource settings, provider or parent perspectives and experiences with CPAP, cost-benefit analyses, and cost-effectiveness studies were included. Inductive content analysis was conducted.

### Findings

1385 article were screened and 54 studies across 19 countries met inclusion criteria. Six major themes emerged: device attributes, patient experiences, parent experiences, provider

**Data Availability Statement:** All relevant data are within the manuscript and its Supporting information files.

**Funding:** The author(s) received no specific funding for this work.

**Competing interests:** At the time of submission, SD, HA, AS, VR, and TB were affiliated with the Vayu Global Health Foundation, which implements novel bubble CPAP systems in low-resource settings. This does not alter our adherence to PLOS ONE policies on sharing data and materials.

experiences, barriers, and facilitators. Nasal trauma was the most commonly reported complication. Barriers included unreliable electricity and lack of bioengineering support. Facilitators included training, mentorship and empowerment of healthcare providers. Device design, supply chain infrastructure, and training models were imperative to the adoption and sustainability of CPAP.

## Conclusion

Sustainable implementation of CPAP in low resource settings requires easy-to-use devices, ready access to consumables, and holistic, user-driven training. Further research is necessary on standardizing metrics, interventions that support optimal provider performance, and conditions needed for successful long-term health system integration.

## Introduction

The World Health Organization has declared the reduction of neonatal mortality a global priority [1]. Each year, two and a half million infants die in their first month of life and the majority of these deaths occur in low resource settings [2]. While considerable progress has been made over the last few decades, respiratory distress syndrome (RDS) remains a leading cause of neonatal mortality worldwide [1–4]. RDS usually develops in the first 24 hours after birth in premature newborns due to a lack of surfactant within the lungs, and often requires positive pressure ventilation for treatment [5]. Continuous positive airway pressure (CPAP) is considered to be the gold standard, treatment for preterm neonates experiencing RDS and is recommended by WHO [6–9].

Forms of CPAP can vary across a number of factors including the patient interface, sophistication, and how they generate pressure. Bubble continuous positive airway pressure (bCPAP) is a common mode of CPAP delivery for newborns that uses a bubbler instead of a ventilator to generate pressure [6–8]. Since bCPAP systems are considered at least as efficacious and are considerably lower cost than ventilator-derived CPAP devices, they may have significant potential to improve access to non-invasive ventilation in low-resource regions worldwide [7, 10, 11]. While reviews of all forms of CPAP [12, 13] have described the efficacy of the treatment, there has been a specific focus on bCPAP therapies suggesting that bCPAP may be safe and effective in low and middle income countries (LMICs) [14–16]. These reviews called for further research on effectiveness and sustainability of bCPAP therapy in low-resource settings [13–16]. A recent systematic review on barriers and facilitators to implementation of neonatal bCPAP among health facilities in sub-Saharan Africa found that staffing ratios, provider knowledge, and device maintenance were crucial to the success of the intervention [17]. However, more information is needed to understand optimization and guide further implementation of all forms of CPAP, including bCPAP, across low-resource settings. Consideration of implementation factors such as successful CPAP device attributes, provider and parent acceptance, and systems uptake must be better understood. Additionally, a broader picture that considers qualitative factors is needed to understand how to create lasting sustainable uptake of CPAP. To explore these factors the following research question was formulated: What are identified barriers, facilitators, and priorities for future implementation of CPAP for neonates and infants in low-resource settings? To answer this more qualitative and nuanced question, a scoping review was chosen to broadly map knowledge gaps and evidence [18].

## Methods

### Search strategy

The scoping review framework was adopted in order to present an overview of all the evidence relating to experiences with CPAP implementation [19]. A scoping review protocol was developed according to the Joanna Briggs Reviewer's manual [20] and this review is reported in compliance with the Preferred Reporting Items for Systematic Reviews and Meta-analyses extension for Scoping Reviews (PRISMA-ScR) checklist (S1 File) [21]. The final protocol was registered on Open Science Framework (https://osf.io/qwvgs/). The search query (S2 File) was run on six databases (MEDLINE, Embase, Web of Science, CINAHL, Global Health, and the WHO Global Index Medicus) from database inception to March 6th, 2020.

### Selection of studies

Search results were uploaded to an online program (Covidence, Veritas Health Information, Melbourne, Australia) to allow for collaborative screening by multiple authors. Four reviewers (SD, RS, HA, AS) independently screened a sample of ten titles and abstracts and agreed on criteria for inclusion and exclusion. Two blinded reviewers (SD, RS) independently screened all articles by title and abstract. Conflicts were resolved by an independent arbiter (RH). Two blinded reviewers (SD, RS) then screened articles by full text for potential eligibility. A final arbiter (RH) resolved conflicts of agreement on inclusion for the final dataset. Original peer-reviewed research articles of any study design on implementation of CPAP devices in low-resource settings as defined by the World Bank Classification at time of study, provider or caregiver perspectives and experiences with CPAP, and cost-benefit analyses or cost-effectiveness studies were included. Grey literature, reviews, and research articles that solely focused on safety and efficacy of CPAP were excluded.

### Data extraction

Three reviewers (SD, HA, AS) independently extracted data from each study using the Covidence data extraction form. Extracted data included: study year; study type/method and setting; population; sample size and method; study objectives; characteristics of CPAP intervention or treatment; complications, barriers, and facilitators. Findings were coded into broad themes by two independent reviewers (SD, HA) using an inductive content analysis on NVivo 12 (QSR International, Melbourne, Australia). An inductive analysis was used in order to uncover patterns and themes in the experiences and perceptions of CPAP implementation [22, 23]. Once all studies were uploaded into NVivo, the two reviewers coded a sample of the studies until data saturation was reached. The individual codebooks were compared and discussed in order to create a final codebook which was then applied to the full dataset.

### Synthesis of results

Studies were grouped by intervention. Broad categories were developed from extracted data related to experiences with implementation of CPAP treatments and results were synthesized across articles. Due to the high variation in study designs and in order to capture and present all of the existing data, studies were not excluded based on quality; and therefore, critical appraisals were not conducted.

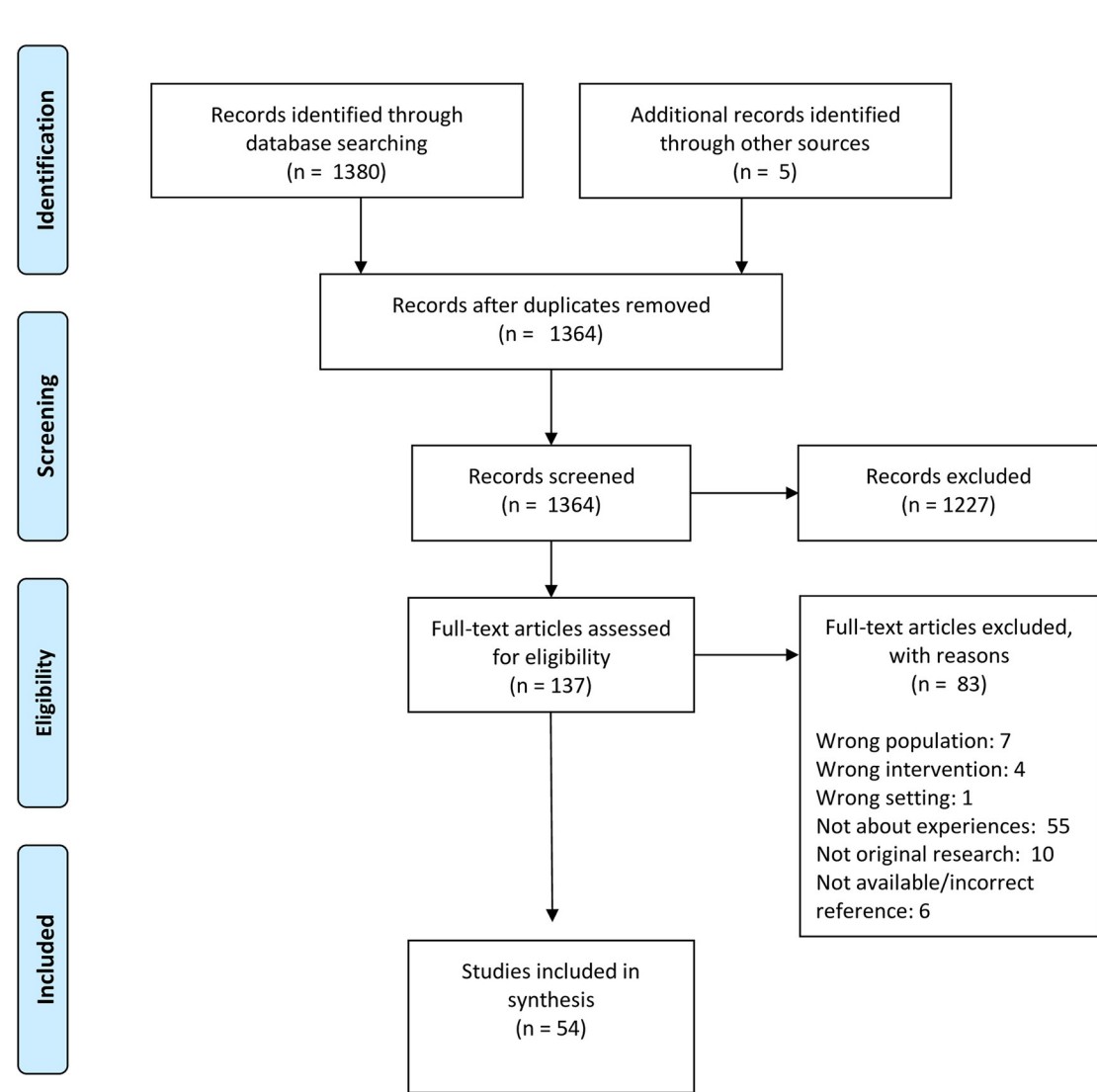

**Fig 1. PRISMA chart.** Adapted from Moher D, Liberati A, Tetzlaff, J, Altman DG, The PRISMA GROUP (2009) Studies included in synthesis (n = 54). *From*: Moher D, Liberati A, Tetzlaff J, Altman DG, The PRISMA Group (2009). *Preferred Reporting Items for Systematic Reviews and Meta-Analyses: The PRISMA Statement.* PLoS Med 6(7): e1000097. doi:10.1371/journal.pmed1000097 **For more information, visit** www.prisma-statement.org.

# Results

## Overview of included studies

Of the 1385 identified studies, 54 were included in the final analysis (Fig 1—**PRISMA chart**) [24]. Included studies are summarized in Table 1. Reasons for exclusion during full-text screening were: incorrect population, unrelated intervention, inappropriate setting, not about experiences with implementation, not original research, unavailable or incorrect reference. No

**Table 1. Summary table of included studies.**

| Study | Country | Specific Intervention | Design | Study participants | Number of participants | Facility Type |
|---|---|---|---|---|---|---|
| CPAP | | | | | | |
| **Abdulkadir 2013** [81] | Nigeria | bCPAP | Case Study | Neonates | 1 | Teaching Hospital |
| **Abdulkadir 2015** [82] | Nigeria | Nasal bCPAP (Improvised) | Descriptive Observational | Neonates | 20 | Teaching Hospital |
| **Al-Lawama 2019** [62] | Jordan | Nasal bCPAP (Fisher & Paykel) | Prospective Observational | Neonates | 143 | Tertiary Care Hospital |
| **Amadi 2019** [25] | Nigeria | Polite bCPAP | Prospective Cohort | Neonates | 57 | Tertiary Care Hospital |
| | | bCPAP (Improvised) | | | | |
| **Antunes 2010** [55] | Brazil | Questionnaire | Descriptive Observational | Nurses | 11 | Tertiary Care Hospital |
| **Atreya 2018** [26] | India | bCPAP (Fisher & Paykel) | Qualitative Interviews | Healthcare providers | 14 | Tertiary Care Hospital |
| **Audu 2014** [27] | Nigeria | bCPAP (Improvised) | Descriptive Observational | Neonates and infants | 48 | Tertiary Care Hospital |
| **Bahman-Bijari 2011** [30] | Iran | bCPAP (Fisher & Paykel) | Randomized Controlled Trial | Preterm neonates | 50 | Tertiary Care Hospital |
| | | vCPAP (Bear Medical Systems) | | | | |
| **Bassiouny 1994** [47] | Oman | Nasal bCPAP (Beneveniste's pediatric gas jet) | Retrospective Descriptive | Neonates | 44 | Teaching Hospital |
| **Boo 2016** [72] | Malaysia | EnCPAP | Retrospective Cohort | Hospital facilities | 34 | Not Specified |
| | | EnCPAP | | VLBW Neonates | 2836 | |
| **Carns 2019** [42] | Malawi | CPAP (Pumani) | Descriptive Observational | Neonates | 2850 | District Hospital |
| **Chen 2014** [28] | Malawi | bCPAP (Pumani) | Cost-Effectiveness Analysis | Neonates | 87 | Not Specified |
| | | Nasal oxygen | | | | |
| **Crehan 2018** [64] | Malawi | bCPAP TRY algorithm (Pumani) | Descriptive Observational | Infants | 57 | District Hospital |
| **Daga 2014** [29] | India | nCPAP (Improvised) | Retrospective Cohort | Neonates | 140 | Teaching Hospital |
| **Dai 2020** [83] | China | nCPAP (CareFusion Infant Flow System) | Prospective Observational | Neonates | 429 | Tertiary Care Hospital |
| **deSiqueira 2014** [84] | Brazil | CPAP | Survey | Nurses | 20 | Teaching Hospital |
| **Dewez 2018** [34] | India | CPAP | Qualitative Interviews | Healthcare providers | 69 | District Hospitals and Medical Colleges |
| **Dewez 2020** [38] | India | CPAP (Improvised) | Cross-sectional Cluster | Hospital facilities | 694 | Government Hospital |
| | | CPAP (Commercial) | | | | |
| **Garcia Reza 2018** [39] | Mexico | nCPAP | Descriptive Observational | Nurses | 25 | Tertiary Care Hospital |
| **Ghorbani 2013** [59] | Iran | nCPAP | Cross-Over Cohort | Preterm neonates | 44 | Teaching Hospital |
| **Gondwe 2017** [61] | Malawi | bCPAP (Pumani) | Qualitative Interviews | Caregivers | 12 | Tertiary Care Hospital |
| **Guedes 2019** [65] | Brazil | nCPAP | Qualitative Interviews | Nurses | 30 | Teaching Hospital |
| **Hendriks 2014** [31] | South Africa | nCPAP (Fisher & Paykel) | Retrospective Descriptive | Neonates | 128 | Rural District Hospital |
| **Hundalani 2015** [66] | Malawi | bCPAP TRY algorithm (Pumani) | Prospective Cohort | Neonates | 325 | Tertiary Care Hospital |
| | | bCPAP early algorithm (Pumani) | | | | |
| | | Oxygen only | | | | |
| **Jardine 2015** [44] | South Africa | bCPAP (Fisher & Paykel) | Retrospective Descriptive | Neonates | 711 | Tertiary Care Hospital |
| **Khan 2017** [45] | India | CPAP (Fisher & Paykel) | Randomized Controlled Trial | Preterm neonates | 170 | Tertiary Care Hospital |
| | | CPAP (Phoenix Medical) | | | | |

*(Continued)*

**Table 1.** (*Continued*)

| Study | Country | Specific Intervention | Design | Study participants | Number of participants | Facility Type |
|---|---|---|---|---|---|---|
| **Koyamaibole 2006** [35] | Fiji | bCPAP (Fisher & Paykel) | Retrospective Cohort | Neonates | 1152 | Tertiary Care Hospital |
| **Myhre 2016** [69] | Kenya | bCPAP (Improvised) | Retrospective Descriptive | Preterm neonates | 118 | Rural Tertiary Care Hospital |
| **Nahimana 2015** [70] | Rwanda | bCPAP (Pumani) | Retrospective Cohort | Preterm VLBW Neonates | 135 | Rural District Hospital |
| **Nyondo-Mipando 2020** [49] | Malawi | bCPAP | Qualitative Interviews | Healthcare providers | 46 | Secondary and Tertiary Care Hospitals |
| **Okonkwo 2016** [71] | Nigeria | bCPAP | Survey | healthcare providers | 237 | Tertiary Care Hospital |
| **Osman 2014** [58] | Egypt | nCPAP | Prospective Cohort | Preterm infants | 60 | Tertiary Care Hospital |
| | | High flow nasal canula | | | | |
| **Sessions 2019** [33] | Malawi | bCPAP | Observational: Time Motion Study | | 12 | Rural District Hospital |
| **Silva 2010** [37] | Brazil | Questionnaire | Qualitative Interviews | Nurses and nursing technicians | 30 | Tertiary Care Hospital |
| **Tagare 2010** [7] | India | bCPAP (Fisher & Paykel) | Randomized Controlled Trial | Preterm neonates | 30 | Tertiary Care Hospital |
| | | vCPAP (Bear Medical Systems) | | | | |
| **Van den Heuvel 2011** [41] | Malawi | bCPAP (Improvised) | Prospective Cohort | Neonates | 5 | Tertiary Care Hospital |
| **CPAP & Training** | | | | | | |
| **Ntigurirwa 2017** [67] | Rwanda | Neonatal training program (427 days) | Retrospective Descriptive | Hospital facilities | 4 | Teaching and District Hospitals |
| | | bCPAP (Fisher & Paykel) | | Infants | 365 | |
| **Olayo 2019** [43] | Kenya | bCPAP training (2 days) | Prospective Cohort | Healthcare providers | 79 | Level 4 and Level 5 Hospitals |
| | | bCPAP (Devilbiss IntelliPAP) | | Neonates and infants | 1111 | |
| **Chen 2017** [63] | Taiwan | Mobile Cart Training | Pre-Post Intervention | Healthcare providers | 59 | Tertiary Care Hospital |
| | | bCPAP (Infant Star v Drager) | | Infants | 113 | |
| **McAdams 2015** [36] | Uganda | RSS Scoring Training | Descriptive Observational | Healthcare providers | 19 | Rural Tertiary Care Hospital |
| | | bCPAP (Improvised) | | Neonates | 21 | |
| **Training** | | | | | | |
| **Asibon 2019** [68] | Malawi | Peer mentorship and training program | Pre-Post Intervention | Nurses | 113 | Secondary and Tertiary Care Hospitals |
| **Tiryaki 2016** [73] | Turkey | bCPAP Lecture | Pre-Post Intervention | Nurses | 36 | University, State and Private Hospitals |
| **Wilson 2014** [32] | Ghana | 1st generation international trainers | Descriptive Observational | Healthcare providers | 28 | District Hospital |
| | | 2nd generation local trainers | | | | |
| **Patient Interface** | | | | | | |
| **Bashir 2019** [53] | India | CPAP nasal mask (Fisher & Paykel) | Randomized Controlled Trial | Preterm neonates | 175 | Tertiary Care Hospital |
| | | CPAP nasal prongs (Fisher & Paykel) | | | | |
| | | CPAP rotating group—prongs and mask (Fisher & Paykel) | | | | |
| **Bonfim 2014** [48] | Brazil | New nasal prongs | Prospective Cohort | Infants with GA < 37 weeks | 70 | Tertiary Care Hospital |
| | | Reused nasal prongs | | | | |
| **Goel 2015** [52] | India | bCPAP prongs (Fisher & Paykel) | Randomized Controlled Trial | Preterm neonates | 118 | Tertiary Care Hospital |
| | | bCPAP mask (Fisher & Paykel) | | | | |
| **Singh 2017** [46] | India | nCPAP nasal mask | Randomized Controlled Trial | Neonates | 75 | Tertiary Care Hospital |
| | | nCPAP nasal prongs | | | | |

(*Continued*)

**Table 1.** (Continued)

| Study | Country | Specific Intervention | Design | Study participants | Number of participants | Facility Type |
|---|---|---|---|---|---|---|
| **Yong 2005** [40] | Malaysia | bCPAP nasal prongs | Randomized Controlled Trial | VLBW infants | 89 | Tertiary Care Hospital |
| | | bCPAP nasal mask | | | | |
| **Nasal Protection** | | | | | | |
| **Nunes 2012** [57] | Brazil | Nasal protection | Case Study | VLBW neonates | 1 | Tertiary Care Hospital |
| **Xiaoyan 2013** [50] | China | Hydrocolloid | Randomized Controlled Trial | Neonates | 500 | Not Specified |
| | | Rhinobyon | | | | |
| **Xie 2014** [51] | China | Hydrocolloid Dressing (Hamilton Medical) | Randomized Controlled Trial | Preterm neonates | 65 | Tertiary Care Hospital |
| | | Paraffin Oil | | | | |
| **Body Position** | | | | | | |
| **Brunherotti 2015** [60] | Brazil | Body position | Cross-Over Cohort | Preterm neonates | 16 | Tertiary Care Hospital |
| **Jabraeili 2018** [56] | Iran | Fetal Position nCPAP | Cross-Over Cohort | Preterm neonates | 50 | Tertiary Care Hospital |
| | | Supine Position nCPAP | | | | |
| | | Prone Position nCPAP | | | | |
| **Pain** | | | | | | |
| **Antunes 2013** [54] | Brazil | Non-nutritive sucking | Randomized controlled trial | Preterm infants | 20 | Government Hospital |

studies were excluded based on language (six non-English papers were translated using Google Translate). Findings were coded into six main categories: device attributes, patient experiences, provider experiences, parent experiences, barriers, and facilitators (Inter-rater reliability kappa score 0.91).

## Description of included studies

The 54 included studies were conducted in 19 countries over five regions: Africa (n = 23), Asia (n = 15), Central & South America (n = 9), Middle East (n = 6), and Oceania (n = 11). Studies ranged from analysis of CPAP treatments (n = 34), training processes (n = 7), patient interfaces (n = 5), nasal protection (n = 3), body positions (n = 2), pain relief (n = 1), and general knowledge or perception surveys (n = 2). Most included studies were randomized controlled trials (n = 10), followed by observational (n = 8) and prospective cohort (n = 6) studies. The most common study populations were term and preterm neonates (n = 18), followed by only preterm neonates (n = 11), and healthcare providers (n = 15). Four studies described their sample population with the general term "infants," which refers to ages 1–12 months, so unless specifically mentioned, the following findings refer to preterm and term neonates, defined as under one month of age.

## Device attributes

Fourteen different CPAP devices were described across the included studies, including Fisher & Paykel (n = 14), Pumani (n = 8) and locally-made or improvised devices (n = 9). CPAP devices varied in price, features, and patient interfaces.

Price was one of the most common themes overall. Five studies emphasized that affordability and cost-effectiveness of different CPAP devices encouraged implementation [25–29] while five studies cited that if a CPAP device was expensive, cost was a barrier to implementation [25–27, 30, 31]. Commercial CPAP devices were noted to have other challenges. For example,

one study reported that nurses found certain CPAP systems "*cumbersome [to set up]*, *particularly securing the tubing to the headdress*" [32]. Sessions et al. measured the length of time healthcare providers (HCPs) spent initiating and monitoring treatment with Fisher & Paykel bCPAP devices and reported it took 12.45 additional minutes to set up and adjust bCPAP equipment compared to the application of standard nasal oxygen [33]. A major focus of most bCPAP devices is to blend pure oxygen with air in order to decrease risk of potential complications from high concentrations of oxygen such as retinopathy of prematurity. However, this complex process is not possible in improvised CPAP devices, and was reported as an important challenge [27].

Important characteristics of various CPAP devices described across the studies included ease of use [25–27, 29, 34, 35] and effectiveness [26, 28, 30, 32, 34, 35]. Ease of use referred to experiences around simple set ups or low maintenance CPAP devices, while effectiveness related to a device's overall ability to provide quality care. Factors such as "*simplicity*" 27] of a CPAP device and "*the feedback provided with use of bCPAP*, *in terms of bubbling of the water column and wiggling of the chest wall*" 26] were cited examples of ease of use. An additional identified device benefit was the potential for certain CPAP devices to be transportable, which could enable use in critical pre-hospital and transit settings [25].

## Patient experience

Twenty-seven studies examined CPAP-related complications and comfort. The most common reported complications were related to nasal irritation [36, 37], nasal lesions [38–40] and abrasions [41] as well as nasal trauma or injuries such as nasal bleeds or hyperemia [40, 42, 43], and nasal septal necrosis [37, 44–46]. Low patient birthweights, low gestational ages [46, 47], and longer treatment times [40, 48, 49] were associated with increased nasal trauma. A number of studies also reported on techniques to reduce nasal trauma through application of protective dressings and use of various patient-device interfaces. In two studies, hydrocolloid dressings, a soft gel-based dressing, effectively reduced nasal injuries [50, 51]. Two of the four studies that compared nasal prongs to nasal masks concluded nasal masks were associated with statistically significant lower incidences of nasal injuries [(36% vs 58%] [52] (33% vs 92%) [53]].

Seven studies described pain or discomfort experienced by a patient on CPAP treatment [37, 45, 54–58]. These studies noted different levels of reported pain (assessed using validated pain assessment tools) based on device type and patient position. Khan et al. found that neonates in a local low-cost CPAP (J-CPAP) group had significantly lower average Neonatal-Pain Agitation and Sedation Scores (N-PASS) than those in a Fisher & Paykel bCPAP group [45]. Osman et al. reported higher pain scores in an nCPAP group compared to high flow nasal cannula [58]. Jabraeli et al. compared pain scores across supine, prone, and facilitated tucking (fetal) positions with nCPAP and described that the lowest pain scores were recorded when the neonate was in a fetal position [56]. Two additional studies found that when neonates received CPAP in a prone position, heart rates and respiratory rates were lower [59], but there were higher rates of nasal prong displacement (56% required repositioning) [60].

## Parent experience

Four studies reported on parents' experience when their newborns underwent CPAP treatment [33, 49, 50, 61]. These studies emphasized that communication between HCPs and parents is important. Parents should be taught about CPAP and engaged in their neonate's care [61]. Two studies described parents' fears related to CPAP treatment [49, 61]. Nyondo-Mipando et al. stated: "*Study participants reported that caregivers sometimes had fears that the many tubes interfered with breathing and that oxygen therapy was associated with death–a*

*perception that may have been influenced by the lack of clear, effective communication between providers and caregivers"* [49]. These two studies also reported on parent interactions with their babies while on CPAP treatment. Participation in their infant's care, such as checking for bubbling in the device, was associated with decreased anxiety and worry [49, 61].

## Provider experience

Multiple studies discussed providers' knowledge of CPAP, device assembly, and patient selection for CPAP treatment. HCPs were more confident in their ability to use CPAP when the devices were simple and accompanied by quality training [26, 32, 34]. Several studies described nurses' perceptions with CPAP treatment [26, 34, 35, 41, 43, 62]. Dewez et al. highlighted *"most nurses felt that trained nurses could initiate CPAP 'independently'"* [34] and Atreya et al. stated that a CPAP device provided *"neonatal nurses with more autonomy"* [26]. In settings with limited medical personnel, this allowed nurses to play an important role in patient care [34].

Six studies described providers' experiences with setting up CPAP devices and initiation of CPAP treatment [33, 39, 63–66]. Nasal prong dislodgment and the need to re-adjust the patient-device interface were common technical challenges during treatment for neonates and infants [7, 45, 67]. Ntigurirwa et al. described these challenges were difficult to address, *"when the nurse to patient ratio is so much lower"* [67]. Additionally, Sessions et al. reported that health workers *"spent an average of 34.71 min longer per patient, initiating bCPAP compared to low-flow oxygen. . . [and] performed, on average, 26.40 more unique tasks"* [33]. Chen et al. addressed this issue by demonstrating that both preparation and application time decreased significantly after staff were trained on a specific CPAP set up protocol [63].

## Barriers

The primary barriers to CPAP implementation were a lack of HCPs and insufficient facility resources. HCP turnover and scarcity were often cited as limitations to effective training and quality patient monitoring [26, 32, 34, 41, 42, 44, 49, 65, 67–70]. Nahimana et al. suggested that gaps in *"correct identification and initiation of eligible infants. . . might be a result of turnover of nurses and doctors"* [70]. A lack of knowledge on how and when to initiate CPAP treatment was another commonly described barrier [26, 34, 37, 41, 49, 63, 70]. One study reported that a lack of device familiarity led to hesitation in use [41]. A lack of familiarity with CPAP may be associated with insufficient staff training [32, 49, 63, 64, 68, 71]. Two studies reported on nurses' hesitation because they were *"afraid of harming neonates because of the need to reuse consumables"* [34] or due to *"fear that the clinician would question their decision"* [49] to initiate CPAP treatment. Other barriers to use of CPAP included lack of institutional buy-in [34, 41] and low staff motivation [67].

Facility resource constraints included lack of uninterrupted electricity, compressed air, oxygen blenders, specific CPAP protocols [72], and computers for record keeping [65]. Reliable electricity was the most frequently described facility infrastructure barrier that affected both patient care [34, 42, 49] and training [68]. In some instances, facility backup generators were not reliable during power outages [49]. Equipment shortages at medical facilities and in supply chains were the most commonly noted of all physical barriers [27, 32, 34, 49, 71, 72]. Amadi et al. identified *"the high cost of devices, consumables and maintenance as limitations to the use of commercial CPAP systems"* [25]. Four studies described it is critical that CPAP replacement parts are available in local supply chains [25, 27, 32, 42]. One study reported that facilities lacked CPAP devices because there were *"not enough machines or many machines were broken"*

[34]. To address these challenges, Carns et al. described that *"spare parts should be easily sourced, and consumables should not be costly"* [42].

## Facilitators

Quality training and mentorship were the most commonly described facilitators for successful CPAP implementation [32, 35, 36, 41–43, 49, 63, 67–71, 73]. Four papers reported that refresher trainings improve CPAP use [32, 49, 68, 70]. Carns et al. described that follow-up *"mentoring visits have ensured continued use of CPAP"* [42] and Ntigurirwa et al. stated, *"through regular, short visits, intensive training can be delivered and problems dealt with. . . but avoids the potential risk of trainers taking over the clinical care of the babies from local staff"* [67]. While some studies reported that CPAP training increases provider knowledge and awareness [42, 63, 73], the most effective approach to training that enables long-term CPAP implementation is not well understood. Wilson et al. implemented a train-the-trainer model where American providers trained Ghanaian nurses, who then trained their colleagues; the latter of whom scored significantly lower on both knowledge and skills testing [32].

Another facilitator described by six studies was the use of an algorithm to guide optimal selection and treatment of patients [36, 44, 49, 64, 66, 70]. Clinical decision algorithms, such as the TRY algorithm were described as easy to teach and integrate [36] to improve infant and neonate treatment [64, 66]. According to Crehan et al, *"the TRY-CPAP algorithm was helpful in guiding healthcare workers in the safe and appropriate application of low-cost bubble CPAP in a district hospital setting where usually physicians are absent and care is nurse-led"* [64]. Additionally, some studies reported on the need for training on bioengineering support for CPAP devices [36, 42]. Finally, two studies identified buy-in from Ministries of Health and policymakers as critical facilitators to successful implementation [26, 42].

## Discussion

This scoping review examined the literature to identify challenges and priorities of CPAP implementation in low-resource settings. Potential priorities for successful CPAP implementation included ease of CPAP device operation [25–27, 29, 34, 35], low cost [25–27, 30, 31], and reliable supply chain for consumables [25, 27, 32, 42]. Common barriers of CPAP implementation included unreliable electricity [34, 42, 49, 68], insufficient CPAP devices and supporting equipment such as pulse oximeters [27, 32, 34, 49, 71, 72], and lack of bioengineering for CPAP device maintenance and repair [32, 34]. Quality training and mentorship that empowered providers facilitated successful CPAP implementation [32, 35, 36, 41–43, 49, 63, 67–71, 73].

A major finding from this review was that it is essential that CPAP devices are easy to assemble, use, maintain, and have simple bioengineering support [33, 39]. Evidence has shown how devices designed in high resource settings are not sustainable as once they break, there is no bioengineering support to fix them [74, 75]. While CPAP devices have traditionally been designed in high resource settings, the unique contexts of low resource regions need to be considered when implementing CPAP across these settings. For example, the polite bCPAP device was specifically designed after surveying Nigerian HCPs on their preferences. Affordability, transportability, and simplicity were the most essential characteristics [25]. The essential takeaway here is that a device's success is dependent on the users and their settings and therefore it is imperative to involve the target audience in the design and implementation process. Such a human-centered design approach has a greater potential to create sustainable, context-based solutions [76]. Incorporating human-centered design facilitates local ownership of CPAP devices and programs by creating a system that may be more appropriate and sustainable [77].

In addition to engineering devices to match their settings, the sustainability of their consumables must also be considered [74]. It is well understood with any device that without available consumables devices will be unusable and only generate waste. That is why it is essential future interventions go beyond facility introduction of CPAP devices to comprehensive integration into health systems in order to ensure sustainability and scale. This includes engaging local manufactures and supply chains. Another solution includes understanding what components could be safely cleaned as reused. Two studies in this review did so for nasal prongs [48, 78], but there is a need to determine safe and standardized reprocessing procedures that are feasible across facilities with different levels of resources. These factors should also be considered in the initial design of devices as mentioned above [76].

Quality training and mentorship were identified as vital facilitators of successful CPAP implementation [17]. Providers must feel confident, empowered, and knowledgeable about CPAP to support and encourage long-term implementation. There is a need for more evidence on different models of training and mentorship, especially taking into account limitations on staff availability. The findings from this review suggest that training models should be integrated into the flow of work with interval in-service training and simulation. As with device design, the development and implementation of training materials should be co-created with local healthcare provider leaders in the settings where they will be used. This will not only foster engagement, but also further adapt education and use to the particular setting in which it will be used [79].

## Limitations

A limitation of this review was the significant variation in study design across the included studies. By setting out to capture a wide range of experiences, we incorporated studies with varied interventions and outcomes. For example, the subset of papers on complications and interventions associated with nasal injury were challenging to compare with studies that reported on the effectiveness of different CPAP devices.

## Conclusion

Inconsistent parameters and outcomes between studies to-date have prevented meta-analyses [13–16]. The study designs, interventions, and objectives in our included studies were also remarkably diverse. Each of the studies in this review addressed an aspect of CPAP implementation that is important to consider when planning for long-term integration of this treatment. While implementation factors are often addressed separately from efficacy and safety in high-resource settings [80], the breadth of experiences described in this review indicates how these measures must be considered concurrently in low-resource settings. Future effectiveness studies should consider not only the short and medium term population outcomes, but also factors that influence sustained integration of CPAP into health systems. A standardized set of implementation outcomes for future research–common barriers and facilitators to study–could allow for improved data synthesis and guidance on optimal care and future research questions.

Successful implementation and integration of CPAP devices across health systems in low-resource settings require appropriate devices, reliable supply chains to replace consumables, and innovative training models that engage users. Each of these elements have one key connection: they each require a deeper engagement of healthcare workers and health systems using these devices. From start to finish CPAP design and implementation should be driven by the final users and the system in which they operate. Combined, it is the hope that these efforts can empower and promote device use, rather than perpetuate potentially unsustainable implementation processes for CPAP use in low-resource settings.

## Supporting information

**S1 File. PRISMA-ScR checklist.**
(DOCX)

**S2 File. Database search queries.**
(DOCX)

## Acknowledgments

The authors would like to acknowledge Harvard Countway Library for the review services provided by Paul Bain in reviewing and running the search query across databases and importing the citations into Covidence.

## Author Contributions

**Conceptualization:** Sara Dada.

**Data curation:** Sara Dada, Rupam Sharma.

**Formal analysis:** Sara Dada, Henry Ashworth, Alina Sobitschka.

**Methodology:** Sara Dada, Rebecca L. Hamilton.

**Resources:** Sara Dada.

**Supervision:** Thomas Burke.

**Writing – original draft:** Sara Dada, Henry Ashworth, Vanitha Raguveer.

**Writing – review & editing:** Sara Dada, Henry Ashworth, Alina Sobitschka, Vanitha Raguveer, Rebecca L. Hamilton, Thomas Burke.

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
