## [Decision Letter · Decision Letter 0]

2 Mar 2021

PONE-D-20-36892

Experiences with implementation of continuous positive airway pressure for newborns and infants in resource-poor settings: a scoping review

PLOS ONE

Dear Dr. Dada,

Thank you for submitting your manuscript to PLOS ONE and my apologies for not getting back to you in a timely manner. I think we were caught in the middle of the end-of-year holiday period which led to a lack of available and suitable reviewers. But having overcome that I hope you find the attached reviewer comments helpful.

After careful consideration, we feel that it has merit but does not fully meet PLOS ONE’s publication criteria as it currently stands. Therefore, we invite you to submit a revised version of the manuscript that addresses the points raised during the review process.

Please consider all the reviewer comments.  

We look forward to receiving your revised manuscript.

Kind regards,

Jacqueline J. Ho, MB.ChB, MMedSc(ClinEpid), FRCP, FRCPCH, FRCPI

Academic Editor

PLOS ONE

Journal Requirements:

2. Please include a caption for figure 1.

Reviewers' comments:

Reviewer's Responses to Questions

**Comments to the Author**

1. Is the manuscript technically sound, and do the data support the conclusions?

Reviewer #1: Partly

Reviewer #2: Yes

Reviewer #3: Partly

2. Has the statistical analysis been performed appropriately and rigorously? 

Reviewer #1: N/A

Reviewer #2: Yes

Reviewer #3: N/A

3. Have the authors made all data underlying the findings in their manuscript fully available?

Reviewer #1: No

Reviewer #2: Yes

Reviewer #3: Yes

4. Is the manuscript presented in an intelligible fashion and written in standard English?

Reviewer #1: Yes

Reviewer #2: Yes

Reviewer #3: Yes

5. Review Comments to the Author

Reviewer #1: The review is interesting, though there are some concerns regarding methodology and reporting. Overall, please review the PRISMA checklist in detail to make sure all points are covered as it is currently insufficiently reported. There is currently some ambiguity between bubble CPAP and CPAP and the manuscript could be strengthened by more clarity on type. There are some recommendations on how the results section could be tightened as currently there is overlap between the different sub-sections. Furthermore, the unique contribution to the literature and implications for policy and practice can be made more explicit and addition of a conclusion section is strongly recommended as the manuscript currently ends very abruptly. Below are detailed comments that I believe will strengthen the paper.

Abstract

1. Missing why CPAP is of relevance to resource-poor settings. Please start with a line or two of background.

2. Please define acronyms the first time they are mentioned, such as CPAP in line 47.

3. Please clarify what is meant by "original research articles and case studies" - case studies would be original research. Do the authors mean both interventions and observational studies?

4. As the authors note that they have used the PRISMA checklist, more information is needed in the abstract. Please see point 2 of the PRISMA checklist: "Provide a structured summary including, as applicable: background; objectives; data sources; study eligibility criteria, participants, and interventions; study appraisal and synthesis methods; results; limitations; conclusions and implications of key findings; systematic review registration number.

Introduction and methods

5. Reference needed for "each year, two and a half million infants die in their first month of life" line 75

6. As a background, more information is needed on respiratory distress syndrome and why it is important particularly to consider for preterm neonates, since it is mentioned that CPAP is recommended in WHO guidelines for preterm infants. Please also quantify "leading cause".

7. The introduction jumps between CPAP and bubble CPAP and it is not completely clear to the reader on what is the focus of the current review. Please clarify the scope of the review (CPAP in general, bubble CPAP because it is more used in LMIC settings?) and the uniqueness of the current review from the previous reviews (i.e. is the current review an expansion of the systematic review of implementation factors for bubble CPAP in sub-Saharan Africa or was it a wider search for all studies of CPAP, not just bubble CPAP?). Why is this review needed?

8. If the focus is on CPAP in general, please provide some background literature review on the implementation of CPAP in resource-limited settings. Currently, the background is focused on bubble CPAP. Is there no examples of other forms of CPAP implemented in LMIC settings? Though I am doubtful that there are no other forms of CPAP implemented, if this is truly the case, this should definitely be added to the background.

9. In line 88, note that the review was on neonatal bubble CPAP, not CPAP in general.

10. The objectives of the review are currently poorly defined. Is the intervention CPAP or bubble CPAP? From the title and introduction, the review appears limited to neonates but this is not stated in the research question. How is newborn defined? How is resource-poor setting defined? Types of studies included?

11. Please explain the scoping review framework further in the methods. It is unclear what this entails and how this is different from a systematic review framework. There is currently no description on why a scoping review methodology was chosen and what makes it unique from a systematic review.

12. Any changes between the protocol and implementation of the review?

13. The eligibility criteria is missing (though reported as completed under item 6 of the PRISMA checklist). Seems like this is in the protocol but more information needs to be added to the manuscript.

14. As stated in item 11 of the PRISMA checklist, please elaborate more on how the reviewer team defined barriers and facilitators. The research question states that it is examining identified "barriers, facilitators, challenges and priorities" (line 95-96). How is barriers different from challenges? What is meant by priorities? Who's priorities?

15. Please describe grounded theory with references, how it was used in this review and provide a coding tree as a supplementary file.

16. Regarding critical appraisals, it states in the protocol that "Included studies will be assessed for risk of bias/quality assessment according to Joanna Briggs Institute (JBI) Critical Appraisal Tool checklists". However in line 132-133, it states that critical appraisals were not conducted. Please clarify this discrepancy and state the justification for amending the protocol if necessary.

Results

17. How many non-English studies were translated? Please add details about the translation process in the methods - was is translated within the reviewer team? Contracted to a professional translator? Google translated?

18. Were the six main categories of coding determined apriori? If so, please revisit grounded theory approach, which is an inductive approach. Is the kappa score regarding the six main categories or the sub-categories? If for the former, what is the kappa for the sub-categories. What was the analysis methodology in developing the sub-categories? More details of this process is needed in the methods.

19. These six main categories should also be added to the abstract. If apriori, then in the methods. If emerged from the data, then in the results.

20. Please define CPAP devices - were they all bubble CPAP devices or other forms of CPAP?

21. For cost effectiveness vs high cost, did it matter between different types of CPAP devices?

22. Please elaborate on why the inability to blend oxygen was reported to be a challenge, particularly to those unfamiliar to respiratory support systems.

23. Easy of use is further elaborated but more detail on what is found for "effectiveness" is needed. Lines 171-6.

24. Please explain what is hydrocolloid dressings? Line 185-6.

25. Please provide more details on why lower birthweight and lower gestational age was associated with increased nasal trauma - is it because nasal prongs were inappropriately sized for small infants. If so, this should be added to the device attributes subsection.

26. The six main categories should be condensed as the first four discuss aspects of CPAP use with barriers and facilitators described. The final two sections on barriers and facilitators should be reworked into the first four categories. Currently, there is overlap and some repetition.

Discussion

27. Table 2 should be reported as part of the results, not the discussion.

28. Regarding "challenges and priorities" (line 284), it is unclear how challenges are different from barriers (see comment 14. Additionally, the results currently do not report priorities. More clarity overall and reporting in the results is required around priority setting.

29. Line 291-292 "lack of bioengineering for CPAP device maintenance and repair" - please rephrase as it is unclear what is meant by bioengineering. Do the authors mean a locally available bioengineering department or that devices need bioengineering development to reduce maintenance and repair needs. If the latter, please discuss some of the innovative systems like Pumani which have aimed to address these concerns.

30. Line 296 regarding devices are easy to assemble, use and maintain: please highlight here that there are different forms of CPAP devices covered in review. Only the Nigerian version was described but this is a form of bubble CPAP. What about different forms of CPAP overall?

31. Are there any policy implications or recommendations?

32. Line 337-339 regarding the current COVID-19 pandemic reads disjointed from the rest of the manuscript and requires further elaboration on what is meant by a it and a "holistic lens".

33. Line 339-340 is a really important point that needs to be made more explicit. Current impact on neonatal survival is still not adequately determined. What do the authors mean by "true impact"?

Conclusion

34. The manuscript would benefit from a conclusion section as it appears missing right now. What is the take home messages and linkages to the bigger picture of neonatal health that the authors would like to conclude after doing their review? Note that this is item 21 in the PRISMA checklist, which is currently inadequately completed.

Reviewer #2: This is a nice scoping review, the primary aim of which is to provide an overview of the available research evidence without producing a summary answer to a "discrete research question", and instead, focusses on a broad research question (well-defined here) . In that regard, the manuscript fits all the requirements necessary for a successful scoping review. In that regard, there is not much statistical analysis to conduct, other than efficient data summarization. I do have some minor comments:

(a) Page 8, line 150: I do not understand how one can have "n = 10" RCTs, when the total number of studies is 54. Is it 40 (unless, the total can't be 54), or I am mistaken?

(b) Page 8, lines 151-153: Same here; the study populations mentioned do not add to 54.

Reviewer #3: - The authors state that they used a scoping review framework referring to the PRISMA checklist. The PRISMA checklist, however, is a framework for reporting the results of a scoping review. It remains unclear which framework the authors used for conducting the scoping review and which considerations determined their decisions. How did the authors deal with critique and recommendations by other scholars, such as Levac and colleagues? This is also not described in the research protocol referred to in the methods section.

- The authors state that they conducted a systematic search. It remains unclear which quality standards/recommendations were applied for conducting the search. A reference to/description of the used framework is missing.

- The authors did not search Pubmed. Why? Pubmed has a broader scope compared to Medline, for example including ahead of print citations.

- The authors state in the review protocol that included studies were supposed to be assessed for risk of bias/quality assessment. However, they decided not to exclude studies based on quality assessment. On page 7 of the manuscript, the authors state that “due to standard scoping review guidelines and variations in study designs, critical appraisals were not conducted” (here, again, it remains unclear which framework the authors used for conducting the scoping review). However, I don’t believe this is not a valid reason to not conduct critical appraisal – just because it does not conventional for scoping reviews. The authors could have decided to use a different method. The question is whether they believe critical appraisal is necessary, and if not, why.

- The authors state that they used a grounded theory approach for the data extraction. It remains unclear which framework the authors used. Grounded theory approach is used to develop new theory from data. However, this is not what the authors have done. They merely conducted a descriptive analysis.

- They research question is limited to the use of CPAP in resource poor setting. What did the authors do with studies in the search result concerning high-resource countries, and why?

- The use/experiences/implementation of CPAP directly after birth is very different from for example 8 months post partum. This difference is not addressed in the result section. I wonder whether the authors paid attention to the relation between age and the use/experiences/implementation.

- In the discussion section, the authors make a few recommendations, for example regarding the availability of consumables. However, they do not address why consumables are not available and what the steps are to making consumable available in in low-resource setting. How is this discussed amongst scholars and in public debate? The analysis remains rather superficial and misses depth in terms of, for example, critically discussing the results in relation to the contemporary body of knowledge, ethics and equity.

6. PLOS authors have the option to publish the peer review history of their article (what does this mean?). If published, this will include your full peer review and any attached files.

Reviewer #1: **Yes: **Mai-Lei Woo Kinshella

Reviewer #2: No

Reviewer #3: **Yes: **Bahareh Goodarzi, Department of Midwifery Science, AVAG, Amsterdam Public Health Research Institute, Amsterdam UMC,

Vrije Universiteit Amsterdam, Amsterdam, The Netherlands.

---

## [Author Response · Author response to Decision Letter 0]

12 Apr 2021

Dear Editorial Team, 

Thank you for sharing the below reviewer comments on our manuscript entitled “Experiences with implementation of continuous positive airway pressure for neonates and infants in resource-poor settings: a scoping review.” 

We appreciate the opportunity to strengthen our manuscript based on the constructive feedback of the reviewers. To date, the majority of systematic reviews about CPAP in low and middle-income settings focus on the safety and efficacy of treatment. This scoping review aims to fill a gap in the literature by summarizing additional contextual factors that influence implementation. We believe that identifying and addressing these factors are imperative to implementing long-term and sustainable CPAP programs in low-resource settings. In particular, the COVID-19 pandemic has highlighted an increased interest toward developing and strengthening available oxygen therapy and ventilation delivery systems. This scoping review aims to better understand lessons learned and opportunities for improvement in the implementation of CPAP therapy for newborns in respiratory distress. 

After incorporating these constructive comments from the reviewers, we believe this article could be a valuable contribution to PLOS One and its publication could add to current discussions and approaches to the implementation of oxygen therapy and ventilation delivery systems in response to the COVID-19 pandemic. Finally, we would like to thank the reviewers for their incredibly helpful comments and feedback that we believe have strengthened the manuscript.

Please see our responses to the reviewers’ comments below. 

Thank you again for your time and consideration. 

Regards, 

Sara Dada, MSc

Corresponding Author

 

Reviewer #1

The review is interesting, though there are some concerns regarding methodology and reporting. Overall, please review the PRISMA checklist in detail to make sure all points are covered as it is currently insufficiently reported. There is currently some ambiguity between bubble CPAP and CPAP and the manuscript could be strengthened by more clarity on type. There are some recommendations on how the results section could be tightened as currently there is overlap between the different sub-sections. Furthermore, the unique contribution to the literature and implications for policy and practice can be made more explicit and addition of a conclusion section is strongly recommended as the manuscript currently ends very abruptly. Below are detailed comments that I believe will strengthen the paper.

Abstract: 

1. Missing why CPAP is of relevant to resource-poor settings. Please start with a line or two of background. 

RESPONSE: We agree that the abstract is strengthened with more information regarding the context and background prior to stating the objective. We have incorporated this (page 2, lines 26-29). 

2. Please define acronyms the first time they are mentioned, such as CPAP in line 47.

RESPONSE: This has been adjusted (page 2, line 26). 

3. Please clarify what is meant by "original research articles and case studies" - case studies would be original research. Do the authors mean both interventions and observational studies?

RESPONSE: We have omitted “case studies” as it is redundant as the reviewer has described. Any original research articles were considered eligible for inclusion (page 2, line 36). 

4. As the authors note that they have used the PRISMA checklist, more information is needed in the abstract. Please see point 2 of the PRISMA checklist: "Provide a structured summary including, as applicable: background; objectives; data sources; study eligibility criteria, participants, and interventions; study appraisal and synthesis methods; results; limitations; conclusions and implications of key findings; systematic review registration number.

RESPONSE: Thank you to the reviewer for highlighting this oversight. We have incorporated a ‘background’ section in the Abstract to address this comment as well as comment #1 (page 2, lines 26-29; lines 38-39). 

Introduction and Methods:

5. Reference needed for "each year, two and a half million infants die in their first month of life" line 75

RESPONSE: The reference for this sentence and the following were the same so this has now been addressed appropriately (page 4, lines 69-70).

6. As a background, more information is needed on respiratory distress syndrome and why it is important particularly to consider for preterm neonates, since it is mentioned that CPAP is recommended in WHO guidelines for preterm infants. Please also quantify "leading cause."

RESPONSE: We appreciate the reviewers comment to provide additional information on respiratory distress syndrome. We have added this detail in the background on RDS (page 4, lines 72-76) 

7. The introduction jumps between CPAP and bubble CPAP and it is not completely clear to the reader on what is the focus of the current review. Please clarify the scope of the review (CPAP in general, bubble CPAP because it is more used in LMIC settings?) and the uniqueness of the current review from the previous reviews (i.e. is the current review an expansion of the systematic review of implementation factors for bubble CPAP in sub-Saharan Africa or was it a wider search for all studies of CPAP, not just bubble CPAP?). Why is this review needed?

RESPONSE: This review focused on implementation of CPAP in general, but we focus on bCPAP overall due to the advantages described in the introduction (such as the lower cost) and provide additional information in the introduction on bCPAP as it is more often used in LMIC settings. We have clarified this in the introduction by describing that there are different forms of CPAP (page 4, lines 78-79) as well as clarified what this review contributes to the growing discussion (page 5, line 96-101).

8. If the focus is on CPAP in general, please provide some background literature review on the implementation of CPAP in resource- limited settings. Currently, the background is focused on bubble CPAP. Is there no examples of other forms of CPAP implemented in LMIC settings* Though I am doubtful that there are no other forms of CPAP implemented, if this is truly the case, this should definitely be added to the background.

RESPONSE: We appreciate this comment as with comment #7 in order to provide further specificity and clarity to the scope of this review. We have added more introductory information describing how CPAPs are differentiated (page 4, lines 78-81) and added additional citations for the efficacy for CPAP in general (page 4, lines 84-85)

9. In line 88, note that the review was on neonatal bubble CPAP, not CPAP in general.

RESPONSE: This has been clarified to specify “neonatal bubble CPAP” (page 4, line 88).

10. The objectives of the review are currently poorly defined. Is the intervention CPAP or bubble CPAP? From the title and introduction, the review appears limited to neonates but this is not stated in the research question. How is newborn defined? How is resource-poor setting defined? Types of studies included?

RESPONSE: We appreciate the call to further clarify our stated objectives. Per comments #7 and #8, we have further clarified that we reviewed CPAP in general, and that specific focus was given to bCPAP as it is often in low-resource settings and appeared in a majority of our results. We have clarified the question to include the patient population that CPAP would be serving (this is consistent with our protocol). The inclusion/exclusion criteria are described in the methods (page 6, lines 140-145). We originally used the term “newborn” to be a less technical term, but because the included literature and rest of the text use the term “neonate” we have changed this to be consistent. This is defined in the results (page 8, lines 195-197). We have also altered the terminology to refer instead to “low-resource” settings as it is more recognizable. We used the World Bank classifications to determine countries that fit this criterion (page 6, lines 142). 

11. Please explain the scoping review framework further in the methods. It is unclear what this entails and how this is different from a systematic review framework. There is currently no description on why a scoping review methodology was chosen and what makes it unique from a systematic review.

RESPONSE: We have specified the scoping review framework and methodology more clearly in the Methods section as well as the reasoning for conducting a scoping review – to be broader and more inclusive in providing an overview of all the evidence (page 5, lines 109-113). Because we did not have predetermined outcomes that we were looking for and wanted to capture all experiences relating to implementation, a scoping review was more appropriate than a systematic review for this study. 

12. Any changes between the protocol and implementation of the review?

RESPONSE: Thank you to the reviewer for bringing this to our attention. The final updated version of the protocol that was used for the review was not properly uploaded to the registration portal (OSF) – this has now been addressed and the publicly available protocol is the version that was used to guide the implementation of the review. The main changes were largely language edits and the decision to not include critical appraisals. 

13. The eligibility criteria is missing (though reported as completed under item 6 of the PRISMA checklist). Seems like this is in the protocol but more information needs to be added to the manuscript. 

RESPONSE: The eligibility criteria is currently described in the methodology selection of studies (pages 6, lines 140-145). 

14. As stated in item 11 of the PRISMA checklist, please elaborate more on how the reviewer team defined barriers and facilitators. The research question states that it is examining identified "barriers, facilitators, challenges and priorities" (line 95-96). How is barriers different from challenges? What is meant by priorities? Who's priorities? 

RESPONSE: As described in the methods section, an inductive content analysis was conducted. We did not define the barriers and facilitators a priori and coded experiences that the primary studies described as barriers and facilitators into these categories. Additional challenges and potential priorities emerged in the analysis and are described in the discussion as areas future research/implementation work must consider. We have clarified the objectives and question of the study to reflect this (page 2, lines 31-32; page 5, lines 101-105)

15. Please describe grounded theory with references, how it was used in this review and provide a coding tree as a supplementary file. 

RESPONSE: Thank you to the reviewer for bringing this up. Upon further reflection, we have instead described this as an inductive analytical approach because we are not presenting a potential theory as some interpretations of “grounded theory” call for. We have updated this in the methodology and have included relevant references (pages 6-7, lines 151-161).

16. Regarding critical appraisals, it states in the protocol that "Included studies will be assessed for risk of bias/quality assessment according to Joanna Briggs Institute (JBI) Critical Appraisal Tool checklists". However in line 132- 133, it states that critical appraisals were not conducted. Please clarify this discrepancy and state the justification for amending the protocol if necessary. 

RESPONSE: As referenced in the response to Comment #12, we have uploaded the updated final version of the review to the registration portal (OSF). We have also added further explanation as to why we did not conduct critical appraisals (page 7, lines 166-168). 

Results: 

17. How many non-English studies were translated? Please add details about the translation process in the methods - was it translated within the reviewer team? Contracted to a professional translator* Google translated?

RESPONSE: This has been clarified (page 7, lines 176-177). We used a Google Translate script to translate the six non-English papers (1 Korean, 1 Spanish, 4 Portuguese) and had a native Portuguese speaker cross-check the Portuguese translations. 

18. Were the six main categories of coding determined a priori? If so, please revisit grounded theory approach, which is an inductive approach. Is the kappa score regarding the six main categories or the sub-categories? If for the former, what is the kappa for the sub-categories. What was the analysis methodology in developing the sub-categories? More details of this process is needed in the methods.

RESPONSE: The categories were not determined a priori which is why we describe this as an inductive content analysis. The kappa score was for all of the nodes that were coded (categories + subcategories). We had three reviewers independently extract the data from all of the studies (each study was extracted twice and then check by the third). All studies were then uploaded into NVivo and two reviewers coded a sample of the studies and then compared and discussed their codebooks to create a final codebook (categories + subcategories) which was then applied to the full dataset. We have incorporated additional explanation into the methods section (pages 6-7, lines 152-161).

19. These six main categories should also be added to the abstract. If a priori, then in the methods. If emerged from the data, then in the results.

RESPONSE: The six main categories are described in the Abstract under findings (page 2, lines 41-43). 

20. Please define CPAP devices - were they all bubble CPAP devices or other forms of CPAP?

RESPONSE: The CPAP devices used in each study are described in the summary table. One of the major challenges we came across was the varying terminology used by different studies – some specified “bubble CPAP” while others used the term “nasal CPAP” even for devices that are known bCPAP systems. We did not want to alter anything that was described in the studies and so we report them verbatim. 

21. For cost effectiveness vs high cost, did it matter between different types of CPAP devices?

RESPONSE: Overall, cost was described as a barrier and included studies did not provide enough information to determine if or how this differed between the types of devices. We thank the reviewer for pointing out the need to clarify and have done so (page 8, lines 204-206).

22. Please elaborate on why the inability to blend oxygen was reported to be a challenge, particularly to those unfamiliar to respiratory support systems. 

RESPONSE: We thank the reviewer for asking to clarify this nuance for readers less familiar with the topic. We have added language explaining the background and difficulty with improvised devices (page 9, lines 214-217).

23. Ease of use is further elaborated but more detail on what is found for "effectiveness" is needed. Lines 171-6. 

RESPONSE: We have provided further clarification on the difference between ease of use and effectiveness throughout the noted paragraph (page 9, lines 221-223).

24. Please explain what is hydrocolloid dressings? Line 185- 6. 

RESPONSE: We have added further description by describing the material used in hydrocolloid dressings (page 10, lines 244-245). 

25. Please provide more details on why lower birthweight and lower gestational age was associated with increased nasal trauma - is it because nasal prongs were inappropriately sized for small infants. If so, this should be added to the device attributes subsection. 

RESPONSE: This is a very interesting question that we have also considered while reviewing the included studies and preparing this manuscript. However, the papers cited did not describe any concrete examples for why this association existed besides the stipulation that lower gestational age and lower birthweight patients had more fragile skin. Since there was not a specific finding or evidence-based conclusion to suggest the reason for this relationship, we have not been able to include this additional detail in our results. 

26. The six main categories should be condensed as the first four discuss aspects of CPAP use with barriers and facilitators described. The final two sections on barriers and facilitators should be reworked into the first four categories. Currently, there is overlap and some repetition.

RESPONSE: We appreciate the reviewers approach to try and condense findings to make them more digestible. We agree that there is natural overlap as all of these topics have important intersections. We also considered this approach initially, but decided to keep them separate as each of these categories should be considered in implementation. The device attributes should be considered by device development groups and organizations selecting a device for implementation. The overall discussion of barriers and facilitators are important considerations for any group to consider and provide a broad overview. The specific sections on providers, parents, and patients provide a more detailed analysis when engaging with key stakeholders. 

Discussion:

27. Table 2 should be reported as part of the results, not the discussion.

RESPONSE: We originally included Table 2 as an alternative way to visualize and organize the main findings on barriers and facilitators but upon reflection and addressing comments from the reviewers, we have decided to eliminate it. 

28. Regarding "challenges and priorities" (line 284), it is unclear how challenges are different from barriers (see comment 14). Additionally, the results currently do not report priorities. More clarity overall and reporting in the results is required around priority setting.

RESPONSE: Our research question includes mention of “challenges and priorities” because we expected these to be areas that needed evidence to be synthesized in order to understand. We did not come across any previous studies that defined a list of known challenges and priorities for future direction of CPAP implementation. The priorities highlighted in this study come from the inductive content analysis and the discussion emphasizes three priority areas for consideration: user-friendly devices and human-centered design, sustainable supply chains, and training. While not all studies explicitly state or declare priorities, the synthesis of our findings pointed to these specific suggested priority areas. 

29. Line 291-292 "lack of bioengineering for CPAP device maintenance and repair" - please rephrase as it is unclear what is meant by bioengineering. Do the authors mean a locally available bioengineering department or that devices need bioengineering development to reduce maintenance and repair needs. If the latter, please discuss some of the innovative systems like Pumani which have aimed to address these concerns.

RESPONSE: We have expanded this part of the discussion to elaborate on the meaning of bioengineering support (page 15, lines 362-365 & 369-372). The theme of bioengineering support refers to the difficulty of maintenance and repair for devices from high-resource settings, ultimately reducing the sustainability of these devices. Per the reviewer’s question, we do feel that it is important to develop devices with this need in mind, and thus we have included the example of the polite bCPAP which was developed alongside Nigerian HCPs. 

30. Line 296 regarding devices are easy to assemble, use and maintain: please highlight here that there are different forms of CPAP devices covered in review. Only the Nigerian version was described but this is a form of bubble CPAP. What about different forms of CPAP overall?

RESPONSE: In this part of the discussion, we are highlighting the relevance of human-centered design and this was the best example of that point. This is something that could be appropriate and applied to other CPAPs but we did not see that in our included studies. This example describes how HCPs were surveyed and how this informed their “main conceptual constraint of politeCPAP was the requirement for high clinical eﬃciency, i.e. lightweight for easy transport, collapsible for ambulatory use, aﬀordable in low-income settings and powered by mains and battery/solar power for ease of use in remote locations where grid electricity is unavailable.” We have incorporated additional detail in this paragraph to clarify that it isn’t necessarily just about CPAPs that are easy to use, but that they are designed with end users in mind, which therefore has implications for ease of use, sustainability, and overall implementation (pages 15-16, lines 377-404).

31. Are there any policy implications or recommendations?

RESPONSE: Thank you to the reviewer for highlighting this very pragmatic question. In the discussion, we focus on three of the main potential priorities for implementation of CPAP (page 14, line 350) – easy to assemble and use devices, sustainable supply chains, and quality training and mentorship. We summarize this again in the conclusion – highlighting the importance of considering sustainable implementation in future research as well, rather than just intermediate health outcomes. 

32. Line 337-339 regarding the current COVID- 19 pandemic reads disjointed from the rest of the manuscript and requires further elaboration on what is meant by a it and a "holistic lens".

RESPONSE: We agree with the reviewer that this is a bit disjointed from the rest of the manuscript and have taken out this reference to the current pandemic and have restructured the conclusion (pages 16-17, lines 423-443).

33. Line 339-340 is a really important point that needs to be made more explicit. Current impact on neonatal survival is still not adequately determined. What do the authors mean by "true impact"?

RESPONSE: This line has been reworked in the process of adding a more distinct and impactful conclusion (pages 16-17, lines 423-443).

Conclusion:

34. The manuscript would benefit from a conclusion section as it appears missing right now. What is the take home messages and linkages to the bigger picture of neonatal health that the authors would like to conclude after doing their review? Note that this is item 21 in the PRISMA checklist, which is currently inadequately completed

RESPONSE: Thank you to the reviewer for this comment. We have incorporated a more explicit Conclusion section with a heading (pages 16, line 423).

Reviewer #2

This is a nice scoping review, the primary aim of which is to provide an overview of the available research evidence without producing a summary answer to a “discrete research question”, and instead, focuses on a broad research question (well-defined here). In that regard, the manuscript fits all the requirements necessary for a successful scoping review. In that regard, there is not much statistical analysis to conduct, other than efficient data summarization. I do have some minor comments: 

1. Page 8, line 150: I do not understand how one can have “n = 10” RCTs, when the total number of studies is 54. Is it 40 (unless, the total can’t be 54), or am I mistaken?

RESPONSE: In this section we reported the three most common study designs across the 54 studies – 10 were RCTs, 8 were described as observational, and 6 were prospective cohort studies. The other 30 studies used a variety of other methodologies or study designs such as case studies, cost-effective analyses, qualitative interviews. All of the studies and their designs are listed in Table 1 (pages 19-23). 

2. Page 8, lines 151-153: Same here; the study populations mentioned do not add to 54

RESPONSE: This is similar to the study designs described below. In this part of the text we are reporting the three most common study populations – 18 focused on term and preterm neonates, 11 only on preterm neonates, and 15 targeted healthcare providers. The remaining 10 studies targeted “infants” (which could range in exact definition/age range) or other specific populations such as very low birthweight infants or specific gestational ages. All of the studies and their participant categories are listed in Table 1 (pages 19-23). 

Reviewer #3

1. The authors state that they used a scoping review framework referring to the PRISMA checklist. The PRISMA checklist, however, is a framework for reporting the results of a scoping review. It remains unclear which framework the authors used for conducting the scoping review and which considerations determined their decisions. How did the authors deal with critique and recommendations by other scholars, such as Levac and colleagues? This is also not described in the research protocol referred to in the methods section.

RESPONSE: We have clarified in the methodology that we used Chapter 11 of the Joanna Briggs Reviewer’s manual to inform the methodology of this scoping review (page 5, line 111). We believe the response to the below comment #2 also addresses the reviewer’s concerns here. 

2. The authors state that they conducted a systematic search. It remains unclear which quality standards/recommendations were applied for conducting the search. A reference to/description of the used framework is missing.

RESPONSE: We have previously described in the methodology that we carried out this review and reported it in compliance with PRISMA-ScR (http://www.prisma-statement.org/Extensions/ScopingReviews). However, to address the reviewer’s comments we have further clarified the methodology and framework used from the Joanna Briggs Reviewer’s manual Chapter 11 which is based on the Arksey and O’Malley framework (page 5, lines 109-113). 

3. The authors did not search Pubmed. Why? Pubmed has a broader scope compared to Medline, for example including ahead of print citations.

RESPONSE: Thank you to the reviewer for bringing this up. We have consulted with the librarian that assisted us with the search query and database search and he has provided the following additional explanation: The data that is covered in MEDLINE in the sets we searched: Ovid MEDLINE(R) and Epub Ahead of Print, In-Process & Other Non-Indexed Citations, Daily and Versions(R) 1946 to March 06, 2020 (https://www.ovid.com/product-details.901.html) is almost to the day exactly the same as the material in PubMed. While we understand that PubMed is typically broader than the MEDLINE subset, the “MEDLINE” data we searched includes that additional material.

4. The authors state in the review protocol that included studies were supposed to be assessed for risk of bias/quality assessment. However, they decided not to exclude studies based on quality assessment. On page 7 of the manuscript, the authors state that “due to standard scoping review guidelines and variations in study designs, critical appraisals were not conducted” (here, again, it remains unclear which framework the authors used for conducting the scoping review). However, I don‘t believe this is a valid reason to not conduct critical appraisal — just because it does not conventional for scoping reviews. The authors could have decided to use a different method. The question is whether they believe critical appraisal is necessary, and if not, why.

RESPONSE: Thank you to the reviewer for raising this point. We did not believe it was necessary to conduct critical appraisals largely due to the fact that we would not be excluding any studies based on quality. In order to remain broad and encompassing as a scoping review, we wanted to be able to capture and present all the relevant, existing data in the literature. We have specified this in the manuscript methodology as well (page 7, lines 166-168).

5. The authors state that they used a grounded theory approach for the data extraction. It remains unclear which framework the authors used. Grounded theory approach is used to develop new theory from data. However, this is not what the authors have done. They merely conducted a descriptive analysis.

RESPONSE: We originally set out to present these six themes as a set of indicators/categories relevant to consider in CPAP implementation – hence the grounded theory approach. However, upon further reflection and because some interpretations of grounded theory call for a proposed framework or theory to be presented at the end of the analysis, we agree with the reviewer that this was not clear. We have edited the manuscript to explain that we used an inductive content/descriptive analysis and describe the process (pages 6-7, lines 152-161). It is an inductive approach because we did not develop these categories a priori but were developed as we began coding. 

6. They research question is limited to the use of CPAP in resource poor setting. What did the authors do with studies in the search result concerning high-resource countries, and why?

RESPONSE: Our search strategy was developed with the guidance of a senior librarian at Harvard and actually included terms for low/poor-resource settings. As a result of this more specified search strategy, we had very few if any studies in high-resource settings in our original list of records. One study was excluded at full text review because of “wrong setting” (figure 1, PRISMA). We decided to focus on low-resource settings from the start because the experiences with CPAP implementation in high versus low-resource settings are very different and likely not comparable. While CPAP is the assumed standard of care in high-resource settings, low-resource settings may not even have CPAP treatments that are accessible or available. The literature on (and experiences with implementation of) CPAP in low-resource settings is growing and the purpose of this review was to synthesize and understand the varying factors that influence that implementation. The reasoning for focusing on low-resource settings has also been further clarified in the abstract (page 2, lines 26-29) and introduction (pages 4-5, lines 81-101). 

7. The use/experiences/implementation of CPAP directly after birth is very different from for example 8 months postpartum. This difference is not addressed in the result section. I wonder whether the authors paid attention to the relation between age and the use/experiences/implementation.

RESPONSE: We thank the review for the comment as we have tried to clarify this terminology. Only four studies looked at infants compared to the vast number of other studies that looked at term and preterm neonates. We have added language explaining how the results focused on this population (page 8, line 193-197). We have gone through and updated the results section when a paper specifically looked and infants and the findings were pertinent to differences in age (page 11, line 286; page 14 lines 338-339).

8. In the discussion section, the authors make a few recommendations, for example regarding the availability of consumables. However, they do not address why consumables are not available and what the steps are to making consumable available in in low-resource setting. How is this discussed amongst scholars and in public debate? The analysis remains rather superficial and misses depth in terms of, for example, critically discussing the results in relation to the contemporary body of knowledge, ethics and equity.

RESPONSE: We deeply appreciate this comment as it is a crucial finding of this paper that we would like to provide a deep and enriching discussion on. We have revised all three subsections of the discussion to include additional references to previous literature on design, consumables & medical waste, and contextual and engaging education (pages 14-15, lines 362-414). We have also revised the conclusion to focus on these three elements, specifically on how they are related and should be a focus of future work (page 17, lines 436-443).

---

## [Decision Letter · Decision Letter 1]

21 May 2021

Experiences with implementation of continuous positive airway pressure for neonates and infants in resource-poor settings: a scoping review

PONE-D-20-36892R1

Dear Dr. Dada,

We’re pleased to inform you that your manuscript has been judged scientifically suitable for publication and will be formally accepted for publication once it meets all outstanding technical requirements.

Kind regards,

Jacqueline J. Ho, MB.ChB, MMedSc(ClinEpid), FRCP, FRCPCH, FRCPI

Academic Editor

PLOS ONE

Additional Editor Comments (optional):

Reviewers' comments:

Reviewer's Responses to Questions

**Comments to the Author**

1. If the authors have adequately addressed your comments raised in a previous round of review and you feel that this manuscript is now acceptable for publication, you may indicate that here to bypass the “Comments to the Author” section, enter your conflict of interest statement in the “Confidential to Editor” section, and submit your "Accept" recommendation.

Reviewer #1: All comments have been addressed

2. Is the manuscript technically sound, and do the data support the conclusions?

Reviewer #1: Yes

3. Has the statistical analysis been performed appropriately and rigorously? 

Reviewer #1: Yes

4. Have the authors made all data underlying the findings in their manuscript fully available?

Reviewer #1: Yes

5. Is the manuscript presented in an intelligible fashion and written in standard English?

Reviewer #1: Yes

6. Review Comments to the Author

Reviewer #1: The revisions were well done and have strengthened the manuscript. I have no further comments to add.

7. PLOS authors have the option to publish the peer review history of their article (what does this mean?). If published, this will include your full peer review and any attached files.

Reviewer #1: **Yes: **Mai-Lei Woo Kinshella

---

## [Editor Report · Acceptance letter]

3 Jun 2021

PONE-D-20-36892R1 

Experiences with implementation of continuous positive airway pressure for neonates and infants in low-resource settings: a scoping review 

Dear Dr. Dada:

I'm pleased to inform you that your manuscript has been deemed suitable for publication in PLOS ONE. Congratulations! Your manuscript is now with our production department. 

Kind regards, 

on behalf of

Professor Jacqueline J. Ho 

Academic Editor

PLOS ONE